



# Design and Field Campaign Validation of a Multirotor UAV and Optical Particle Counter

Joseph Girdwood[1], Helen Smith[1,a], Warren Stanley[1], Zbigniew Ulanowski[1,b,c], Chris Stopford[1], Charles Chemel[1,2], Konstantinos-Matthaios Doulgeris[3], David Brus[3], David Campbell[4], and Robert Mackenzie[1]

[1]Centre for Atmospheric and Climate Physics, School of Physics, Astronomy and Mathematics, University of Hertfordshire, Hatfield, Hertfordshire, AL10 9AB
[2]National Centre for Atmospheric Science, Centre for Atmospheric and Climate Physics, School of Physics, Astronomy and Mathematics, University of Hertfordshire, Hatfield, Hertfordshire, AL10 9AB
[3]Finnish Meteorological Institute, PO Box 503, FI-00101, Helsinki, Finland
[4]School of Physics, Astronomy and Mathematics, University of Hertfordshire, Hatfield, Hertfordshire, AL10 9AB
[a]Now at: TruLife Optics Ltd, 79 Trinity Buoy Wharf, London, UK
[b]Now at: Centre for Atmospheric Science, University of Manchester, Manchester, UK
[c]Now at: British Antarctic Survey, NERC, Cambridge, UK

**Correspondence:** Joseph Girdwood (j.girdwood@herts.ac.uk)

**Abstract.** Small unmanned aircraft (SUA) have the potential to be used as platforms for the measurement of atmospheric particulates. The use of an SUA platform for these measurements provides benefits such as high manoeuvrability, re-usability, and low-cost when compared with traditional techniques. However, the complex aerodynamics of an SUA (particularly for multirotor airframes), combined with the miniaturisation of particle instruments poses difficulties for accurate and representative

sampling of particulates. The work presented here relies on computational fluid dynamics with Lagrangian particle tracking (CFD-LPT) simulations to influence the design of a bespoke meteorological sampling system: the UH-AeroSAM. This consists of a custom built airframe, designed to reduce sampling artefacts due to the propellers, and a purpose built open-path optical particle counter—the Ruggedised Cloud and Aerosol Sounding System (RCASS). OPC size distribution measurements from the UH-AeroSAM are compared with the Cloud and Aerosol Precipitation Spectrometer (CAPS) for measurements of Stratus

cloud during the Pallas Cloud Experiment (PaCE) in 2019. Good agreement is demonstrated between the two instruments. The integrated dN/dlog(Dp) is shown to have a coefficient of determination of 0.8, and a regression slope of 0.9 when plotted 1:1.

## 1   Introduction

Aerosols and their interactions with clouds and radiation have been consistently highlighted by the Intergovernmental Panel on Climate Change (IPCC) as the largest uncertainty in predicting climate change today (IPCC, 2013). This is, in part, due to

the difficulty of measuring such phenomena, and their high spatial heterogeneity. To reduce this uncertainty, regular measurements of aerosols and droplets (and radiation) would need to be performed with the aim of quantifying the aerosol-radiation and cloud-aerosol interactions that affect our climate. Currently, aerosol measurements are conducted using remote sensing techniques (e.g. sun photometers and LIDARs) on ground based (Bokoye et al., 2002; Che et al., 2009; Baars et al., 2015)





and satellite instruments. While these methods require the least personnel and can sample a continuous vertical column, they directly measure the column-integrated optical parameters of the atmosphere, and require complex algorithms (e.g. LIRIC and GARRLiC Tsekeri et al., 2017) and various assumptions about the atmosphere to retrieve aerosol properties. Hence remote sensing requires validation from coincident, height resolved, in-situ meteorological measurement. Additionally, most remote

sensing techniques have limited vertical resolution, and are unable to measure the lower parts of the planetary boundary layer (PBL). This makes in-situ measurements the only suitable sampling method for the first $100\,\mathrm{m}$ of the atmosphere.

In-situ aerosol and droplet data are conventionally collected using manned aircraft (Hara et al., 2006; Drury et al., 2010) and, to a lesser extent, meteorological soundings (both on tethered and non-tethered balloons) with instruments like the LOAC (Renard et al., 2016, 2018), the UCASS (Smith et al., 2019), and the instrument developed by Gao et al. (2013). Manned aircraft

in particular are extremely expensive to operate, maintain, and crew. Also, vertical profiling is often a valued measurement, not only for remote sensing validation, but also because important atmospheric phenomena—such as moist convection—are governed by processes occurring in the vertical direction. A direct vertical profile cannot be accomplished by a fixed-wing manned aircraft due to restrictions with mobility, often meaning "staircases" or spiral ascents with large lateral dimensions have to be employed instead. Thus, such measurements can be problematic for comparisons with remote sensing in atmospheres with

high spatio-temporal variability. Furthermore, the aircraft based instruments used to measure aerosol and droplets—for example the Forward Scattering Spectrometer Probe (FSSP, evaluated by Dye and Baumgardner, 1984; Baumgardner et al., 1985; Baumgardner and Spowart, 1990); the Cloud Droplet Probe (CDP, described by Lance et al., 2010); and the Backscatter Cloud Probe (BCP, Beswick et al., 2014)—incur huge costs, and availability is often a concern. Meteorological soundings can—to an extent—negate these issues since a vertical profile can be accomplished, and some airspace restrictions—for example runway

availability—do not apply. However, the payload from a non-tethered balloon is not often retrievable, making regular soundings with aerosol instruments to examine temporal variations impractical.

Unmanned aerial vehicles (UAVs), referred to here as small unmanned aircraft (SUA) according to the UK Civil Aviation Authority (CAA) definition, are becoming increasingly popular for PBL research. Fixed wing SUA designed for meteorological sampling (e.g. the designs described in Buschmann et al., 2004; Reuder et al., 2009; Wildmann et al., 2014a; Altstädter

et al., 2015) have been used in numerous studies already, including, but not limited to: new particle formation (Altstädter et al., 2018); the sampling of standard met-sonde parameters (Martin et al., 2011); measuring the wind vector using a 5-hole pitot probe (van den Kroonenberg et al., 2008); the sampling of black carbon aerosol (Ramanathan et al., 2007; Corrigan et al., 2008); Saharan dust aerosol (Mamali et al., 2018); Arctic boundary layer aerosol (Bates et al., 2013); and turbulent flux measurements (Wildmann et al., 2013, 2014b, c). Rotary wing platforms (e.g. multirotors) on the other hand have been

used much less extensively in atmospheric physics, likely due to problems with the validation of measurements—because of stronger aerodynamic distortions—and limitations to their endurance. However, if these issues can be overcome, multirotor platforms present many advantages over fixed-wing based platforms since they can fly directly upwards for a vertical profile; they integrate very well with auto-pilot systems allowing precise flights with minimal human interference; they require less pilot experience to operate effectively; and measurements can be repeated easily in the same location, thus providing superior

spatio-temporal sampling abilities.





However, the scientific validity of quantitative measurements conducted using any platform can be questioned if a proper validation process has not been implemented. This is especially true when sampling atmospheric aerosol and droplets using a multirotor SUA, due to artefacts resulting from the the aerodynamic disturbances created by the propellers. Quantifying this distortion, and its effect on particles, can be difficult due to the complexity of flow measurements (especially when considering

turbulence). Alvarado et al. (2017), for example, presented an approach to validation involving anemometer-based measurements of the air velocity (in the vertical direction) in a grid pattern around a multirotor SUA at full throttle. This approach, however, will not provide enough spatio-temporal resolution for turbulence measurements, and only involves flow measurements vertically. Also, this method does not account for any crosswind, or the motion of the SUA itself with respect to the surrounding air. Another method that is commonly applied to the validation of particle instruments, but could also be applied

to SUA, is wind tunnel testing. Clarke et al. (2002) used a wind tunnel to post-evaluate a miniaturised optical particle counter (OPC), and derive a correction factor for sub-isokinetic sampling flow which can cause an under-prediction in particle concentration measurements. This technique, however, is generally only used to simulate the effects of the flow speed and angle of attack without the SUA (or manned aircraft) airframe. This is because wind tunnels are normally not large enough to accommodate the SUA, and a dimensionally similar scale model cannot be used (a technique common in the aerospace industry)

since the particle instrument would have to be full size.

The purpose of this paper is to present a novel design and validation technique for sampling aerosols on a multirotor SUA. The main justification behind this approach is that the validation is simpler—and the data products more reliable—when artefacts of measurement and experimental design are considered throughout the physical engineering design process. We utilise computational fluid dynamics with Lagrangian particle tracking (CFD-LPT) as a tool to influence design decisions for the

SUA—thus identifying the main sources of measurement error—which were then field tested with reference instrumentation for validation. The aircraft used in this experiment is the aerosol sampling SUA UH-AeroSAM, which is a bespoke SUA developed at the University of Hertfordshire and equipped with an open path OPC to sample atmospheric aerosol and droplets. An overview of some existing studies using SUA to measure atmospheric particulates is presented in Sect. 2, and a description of the UH-AeroSAM configuration and instrumentation is presented in Sect. 3. The CFD-LPT simulations are presented in Sect.

4, and the field validation is presented in Sect. 5. The field validation took place during the Pallas Cloud Experiment (PaCE, 2019)—a biennial experiment at the Pallas atmosphere-ecosystem super-site (Lohila et al., 2015) with the aim of characterising sub-Arctic cloud, and validating instruments.

## 2 Overview of Existing SUA Particle Measurements

While aerosol and droplet measurements on SUA are relatively novel, previous SUA studies have attempted to measure the

physical properties of atmospheric particles and droplets. One such study was Bates et al. (2013), which used the MANTA SUA as a platform for a three-wavelength absorption photometer, a condensation particle counter (CPC), and a chemical filter sampler (all connected to the same artificially aspirated inlet) to measure vertical black carbon (BC) concentrations in the Arctic. This study represents an effort to obtain a greater understanding of (potentially anthropogenic) BC transport into Arctic





regions, and concluded that regular SUA measurements could provide the in-situ BC (and other aerosol) data needed to validate climate models and remote sensing retrievals. However, in-situ methods (particularly using SUA) still require validation and testing before the data can be trusted for these purposes.

Another step towards the reliable, regular use of SUA data in climate models was presented in Mamali et al. (2018). This

study compared vertical profiles of Saharan dust mass concentration measured using an artificially aspirated OPC mounted on a fixed wing SUA, to remote sensing retrievals (POLIPHON) over Cyprus in 2016. While the coefficient of determination between the POLIPHON-retrieved and OPC-derived aerosol mass concentration was found to be 0.8 in the first case study and 0.72 in the second, there are elements of the dust vertical structure (for example sharp changes in the spatial mass concentration profile) that were measured by the SUA but not by the remote sensing. This could be caused by such phenomena as: a highly

heterogeneous dust layer causing a spatial variation in mass concentration between measurements; some measurement artefact resulting from the SUA airframe or OPC; or an underlying problem with the POLIPHON retrieval. Multirotor airframes have better spatio-temporal sampling abilities, thus giving them the ability to fly along a profile co-located to that of the LIDAR. Additionally, proper validation of the airframe and in-situ instrumentation can minimise any artefacts in the SUA data, and a comparison of different retrievals (for instance in Tsekeri et al., 2017) can help discover problems with remote sensing. For

these reasons, more work is still to be done on the proper integration of SUA data into model datasets and remote sensing validation.

SUA enable regular measurements of atmospheric properties that cannot be accomplished by any other platform. This was fully exploited by Altstädter et al. (2018) where the fixed-wing ALADINA SUA initially proposed by Altstädter et al. (2015), using a new setup described by Bärfuss et al. (2018), was utilised for observations of new particle formation (NPF).

This study was a strong example of SUA utilisation where manned aircraft measurements would be impractical. Due to the temporally variable nature of the NPF process, regular measurements would be necessary, which would incur large costs in manned aircraft. Additionally, the NPF events observed here, and during a previous study by Platis et al. (2016), occur at low altitudes where manned aircraft often cannot fly. In the UK for example, manned aircraft are often prohibited from flying below 500 ft where many of these NPF events occur. This study represents a promising step towards the regular measurements

of ultra-fine atmospheric aerosol properties. However, these particles tend to be subject to higher aspiration and transportation losses, especially in turbulent flow. A series of commonly used empirical formulae initially derived from wind tunnel testing is reviewed in Von Der Weiden et al. (2009). These formulae are used to predict the losses due to the aerosol instrument transportation and aspiration mechanism, but not airframe effects which, while not as predominant on a fixed-wing SUA like "ALADINA", would produce artefacts in multirotor measurements. For that reason, a CFD or experimental technique must be

used to characterise airframe related artefacts.





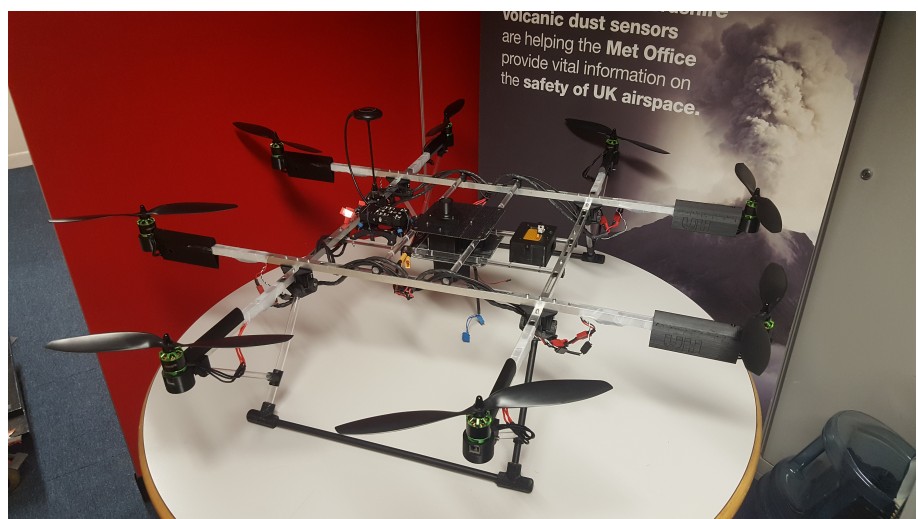

**Figure 1.** The University of Hertfordshire aerosol sampling SUA (UH-AeroSAM). This is a bespoke SUA specifically designed to sample aerosol particles and droplets using a custom built OPC.

## 3 Description of UH-AeroSAM

### 3.1 Airframe and Auxiliary Instrumentation

UH-AeroSAM is a custom built octocopter SUA, with opposing motors separated by approximately 1m. Figure 1 shows an image of UH-AeroSAM with its full instrumentation package installed. The open-geometry frame allows for a highly config-
urable payload. Consequently, the SUA can accommodate additional or different instruments in future studies. The maximum take-off mass (MTOM) is $3.2\,\mathrm{kg}$, although lower take off masses will allow the SUA to achieve a longer endurance, and hence a higher altitude if a vertical profile is desired. UH-AeroSAM is controlled using a Pixhawk flight controller (3DRobotics, 2013) with an external GPS module, which is tuned for stability in wind gusts up to $15\,\mathrm{m\,s^{-1}}$. The positioning of the particle sensor on the airframe is discussed in Sect. 4.2.
UH-AeroSAM is equipped with an adaptation of the Universal Cloud and Aerosol Sounding System (UCASS, original design: Smith et al., 2019), the modified design is presented in Sect. 3.2. This OPC is a ruggedised version of the original UCASS (a single-use instrument), which withstands multiple flights, with high levels of vibration. This re-design also uses an inlet which can be easily swapped, incorporating design optimisation resulting from CFD-LPT. The OPC is also a naturally aspirated—also known as 'open-path'—OPC, which relies on the movement of the platform itself to provide a flow of particles
through a sensing area, as opposed to a fan or pump. The motion of the SUA platform must be taken into account during the CFD-LPT simulations.



For atmospheric research, there exists several benefits and caveats when considering building versus purchasing an airframe. In a changing world with regards to legislation on SUA, purchasing an airframe from an officially recognised distributor appears to be the most legislatively stable option. This, however, restricts the operator on payload configuration and weight— an essential consideration for atmospheric measurement in general, since the positioning of scientific instrumentation on an airframe influences the quality of meteorological data. For this application in particular, an airframe with an open geometry configuration and propellers positioned far away from the centre was not, to the authors knowledge, previously commercially available, because the centre is normally reserved for the flight controller and battery pack. The other option for positioning the OPC away from the propeller effects is to mount a boom extending outwards from the SUA, and attach the OPC on the end. This, however, would result in a disruption to the stability of any standard airframe, thus causing it to be categorised as a 'home-built' airframe which infers more strict legislation by governing bodies in some countries.

Data from the OPC is available through a serial peripheral interface (SPI). This is connected to a data-logger, which is based on a Raspberry Pi zero (RaspberryPi-Foundation, 2015). This records data from 16 configurable size bins with a frequency of $2\,Hz$, where each particle size distribution is integrated over this interval. This was found to be the greatest temporal resolution the data logger could handle, although at an ascent rate of $5\,m\,s^{-1}$ (a typical balloon ascent rate) it gives a better (vertical) spatial resolution than a balloon-based sonde, which often record at 1Hz while using an x-data interface (for example the Graw DFM-09). The GPS data is corrected using an inertial measurement unit (IMU) located on-board the Pixhawk, and transferred to the data-logger using the 'MAVLINK' protocol. Pressure and time data are also transferred to the data-logger from the Pixhawk using this protocol. The time data is synchronised with a real time clock every 0.5 seconds.

Temperature and humidity are also measured on the SUA. A fast-tip glass bead thermistor (FP07 Fastip Probe) and a HIH-4000 capacitive humidity sensor are mounted in a cylindrical radiation shield. The enclosure is positioned underneath the propeller for enhanced aspiration, one quarter the length of the propeller from the tip as recommended by Greene et al. (2018). It was found that the increased pressure in the enclosure due to the propellers had a negligible effect on temperature and humidity measurements. The radiation shield is constructed from P2T (recyclable carbon fibre) with a gold coating around the exterior to reflect solar radiation and avoid radiative heating of the sensor. Since the HIH-4000 sensor contains an exposed silicon element, it has a tendency to act as a photo-diode when exposed to large amounts of stray light, and give saturated humidity measurements. To avoid this, the interior of the tube was coated with a broad-band absorbing paint (Stuart Semple - Black 2.0) so stray light reflected onto the sensor element is minimised.

## 3.2 Aerosol Instrumentation

The OPC used on the SUA is optically and electronically similar to the UCASS. However, since a SUA platform presents entirely different design criteria to the original dropsonde design, a mechanical re-design was necessary. The first principle of design was endurance, since the original UCASS is a single-use instrument and a SUA platform is likely to subject the payload to vibrations not considered in the UCASS. The Ruggedised Cloud and Aerosol Sounding System (RCASS) therefore features an aluminium chassis and a more robust optical alignment system to ensure the optical components do not move. A computer aided design (CAD) model of the RCASS is shown in Fig. 2. Since the original design was influenced with constraints

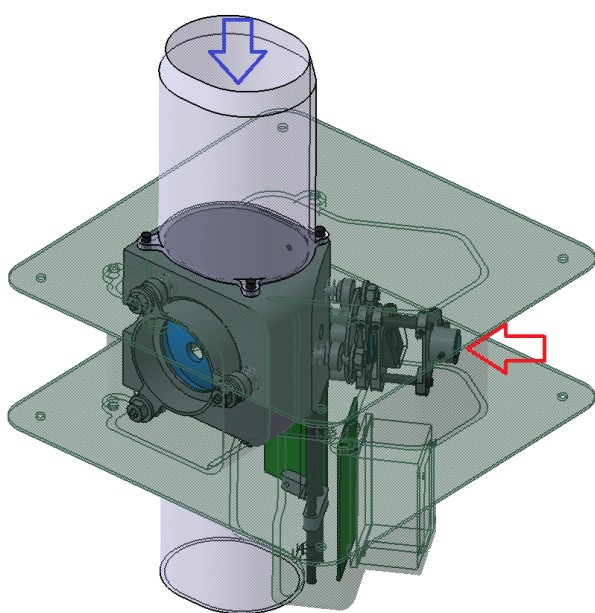

**Figure 2.** The RCASS—a mechanical re-design of the UCASS (Smith et al., 2019). The red and blue arrows are the laser beam and airflow directions respectively.

associated with other combined systems, a mirror placed at a 45°angle directs the laser beam into the sampling tube, which is intended to reduce the package size of the instrument. The same design constraint is not present here. Therefore, in order to simplify construction, and reduce maintenance costs (e.g. optics cleaning), the optical carrier is mounted orthogonally to the sampling tube with no mirror redirecting the beam.

The mechanical re-design of the instrument also allowed an improvement of the internal aerodynamics. Smith et al. (2019) shows CFD results used to determine operational limits for the instrument based on platform-velocity derived mass concentration. However, the CFD simulations of the UCASS show an area of high stagnation pressure at the inlet on which a leading-edge vortex forms, directing particles around the sampling volume, and thus altering the mass flux through the detector region. This effect can be seen in Fig 3 which shows the flow profile through the UCASS for an input angle of attack of $-20°$—this effect

is demonstrable for all angles of attack (and in axial flow), but $-20°$ is shown for the sake of brevity. This is amplified for negative angles of attack and high airspeeds ($>10\,\mathrm{m\,s}^{-1}$), suggesting that the blunt geometry of the leading edge face is causing this high gauge pressure, and therefore the leading edge vortex. Hence the RCASS inlet features a sharp tip similar to the 'Korolev' cloud probe tips presented and simulated in Korolev et al. (2013). Although the Korolev tips are designed to prevent shattered ice particles being sampled, the underlying process directing the ice particles into the sample volume is similar with

a high pressure region forming on blunt tips forcing ice shards into the sample volume. As shown by Jackson et al. (2014), the anti-shattering tips can impact historical data—demonstrating the need to assess artefacts of measurement retroactively.



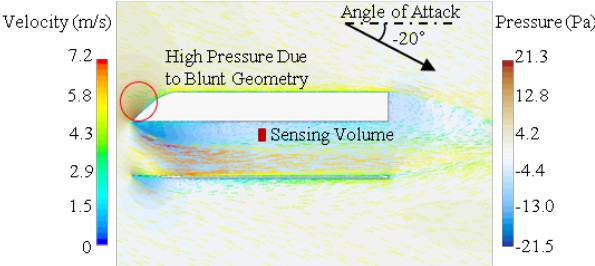

**Figure 3.** A vector plot of CFD simulation results for the original UCASS with an airflow angle of attack of $-20°$. The arrow represents angle of attack, the red circle is the high pressure region which directs particles around the sensing volume, which is labeled as a red square. The scale on the right is gauge pressure—corresponding to the colour of the background; the scale on the left is airflow velocity magnitude—corresponding to the colour of the glyphs.

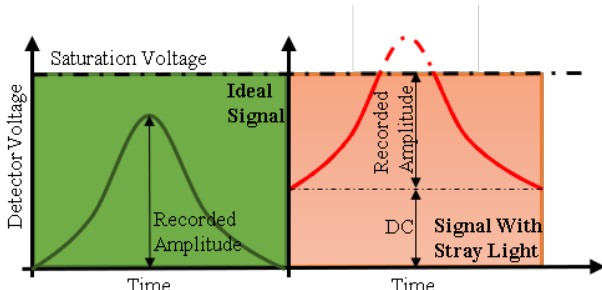

**Figure 4.** An illustration of the effect of stray-light on the signal output of the detector electronics (the transimpedance amplifier). Excess stray-light will cause the Gaussian pulse from a particle to stray beyond the saturation point, represented in the red area of the figure.

Another limitation with the UCASS is the positioning of the detector region in the boundary layer of the instrument, therefore causing particles to be sampled in a region with lowered airflow velocity, and an increased likelihood of turbulent deposition. This is also shown in Fig 3. The new design, therefore, features a sample area positioned further away from the wall of the airflow tube. Following these results, the UCASS was also modified to include a collar around the inlet, which prevented the

5    high pressure region from forming.

   Since the package had been changed, the stray light issue (common among all OPCs, especially open path) needed to be re-addressed. The RCASS electronics can endure a certain amount of stray light using a direct current (DC)-restoration circuit. This is an inverted peak-detector circuit which will hold the value of the DC signal (a result of the stray light on the detector), minus any peaks from detected particles and noise. The DC signal is then subtracted from the total signal in

10   a comparator. However, if the DC component is sufficiently large, the detector can start to saturate, exhibiting—in an ideal circuit—a Gaussian distribution with a flat top. Once the DC signal has been subtracted from this, the peaks appear smaller





in amplitude, leading to the appearance of smaller particles—and eventually nothing—to be recorded. This effect is illustrated in Fig. 4. A stray light test was devised on a prototype RCASS design, which was improved and verified following the tests. These tests are described in detail in appendix B.

The original UCASS is available in two different gain modes: low gain to capture larger droplets ($3\,\mu m$ to $40\,\mu m$); and high
gain to capture smaller droplets ($0.4\,\mu m$ to $20\,\mu m$). On the RCASS, a mechanical switch was installed to allow the operator to change the gain mode before a flight, depending on meteorological conditions.

## 4   Computational Fluid Dynamics with Lagrangian Particle Tracking

### 4.1   CFD-LPT Methods

Simulations were performed to determine the best place on the SUA to position the RCASS. It was hypothesised that the
minimal distortion of OPC measurement would be in the centre of the SUA; however a design compromise existed between the centralisation of the OPC, or the Pixhawk flight controller. A centralised flight controller makes easier dynamic stability tuning, but this may compromise OPC measurements.

Since the particle inlet is above the airframe, the aerodynamic effects of the airframe itself are negligible, and therefore have been ignored. This also allows lower computation and meshing time. The simulation domain was a virtual wind tunnel
consisting of a $4\,m$ tall cylinder with a radius of $1.5\,m$; the eight propellers were positioned in this cylinder on a plane $2.5\,m$ from the base. The centre of each propeller was located on the vertices of an octagon with an 'across-flats' dimension of $0.97\,m$—the same geometry as the real SUA. The walls of the cylinder are assumed to generate no shear stress, and hence no boundary layer. This was to simulate a quasi-free-stream while simultaneously reducing computation time. Figure 5 shows a schematic of the CFD-LPT domain and propellers.

The motion of the propellers was simulated using an overset mesh (also known as a 'Chimera' mesh), which was chosen for its compatibility with the time unsteady solver (required for LPT), and its flexibility with different mesh types.

The transport equations used in the simulations were the Reynolds averaged Navier Stokes (RANS) equations, with a k-ε turbulence model. This is a two-equation eddy viscosity model, which is commonly used for its stability and accuracy in the free stream. The k-ε model tends to under-predict the anisotropy of turbulence near walls and in wakes, however the points of
interest on the airframe were in the far field, away from walls. Tian and Ahmadi (2007) found that a Reynolds stress model (RSM) would more accurately predict near-wall turbulence conditions. However, in complex flows such as this, a RSM would be highly unstable and require minute time steps, which would vastly increase computation cost. While near-wall turbulence strongly influences turbulent deposition of particles, this effect was not expected to be important in this case, since all areas of interest are far away from any boundary layers. Therefore, a k-ε turbulence model was sufficient.

These transport equations were solved numerically by an implicit unsteady solver, which was available in the "Star CCM+" commercial code used. This solver converges a simulation for each time step with multiple inner iterations (sub-steps). The quasi time step for these inner iterations was calculated from a user-defined Courant number which was set to 50, other internal Courant numbers were experimented with but this was the best compromise between stability and computation time. The time-


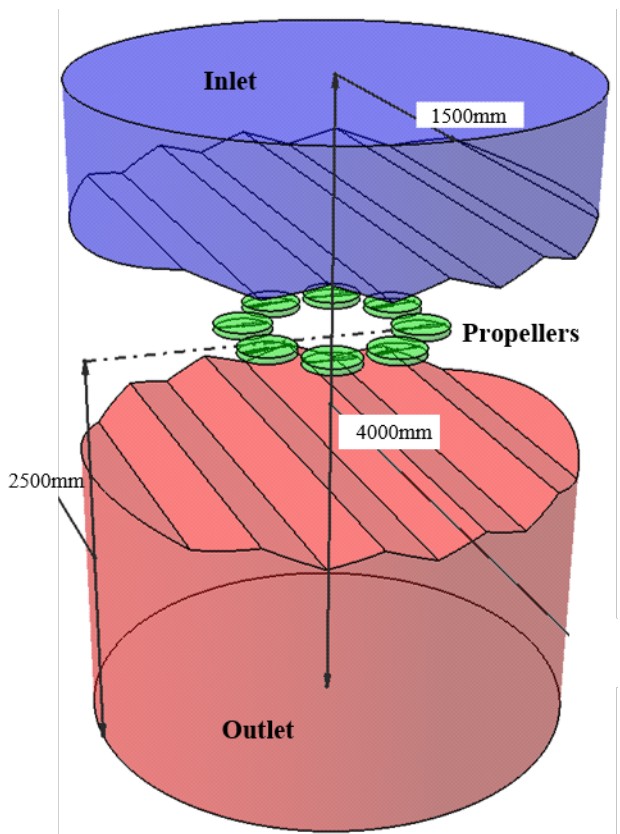

**Figure 5.** This figure shows the CFD-LPT domain. Air and particles enter the domain at the top of the blue section labeled "Inlet", and flow downwards past the propellers. Each of the propellers is in a small green cylinder representing the overset mesh region; a section of the cylinder is cut away here for illustration purposes. Air and particles leave the domain through the bottom face of the cylinder in the red region.

step for all simulations was 0.0005 s, which was chosen to satisfy both the Courant–Friedrichs–Lewy (CFL) condition for the mesh size chosen, and the Nyquist-Shannon sampling criterion for the motion of the propellers. Each simulation was run until it had converged, and the first particles injected into the simulation had left the domain. A sensitivity study was conducted to ensure the precision and reliability of the simulation results. This is presented in appendix A.

5      The LPT component of the simulations used two models: a simple drag model to study the viscous and inertial forces exerted on the particles by the fluid, and a turbulent dispersion model to study the effects of turbulence. Models that effect the particle size distribution—for example droplet breakup and coagulation—are not studied in these simulations due to their complexity and lack of validation. Instead, the droplet Weber number, and fluid Reynolds number were calculated to ensure this did not happen (see Sect. 4.2). Particles in the simulation were 'sampled' when their trajectory during one time-step bisected a plane

10    parallel with the $x$ and $y$ axes of the domain (parallel with the propeller plane shown in Fig. 5). These planes were positioned





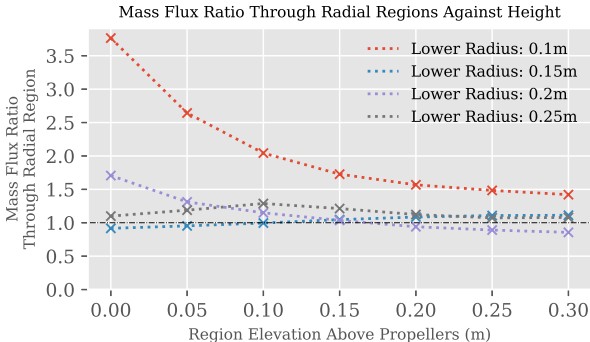

**Figure 6.** CFD-LPT results for the ratio of particle mass flux at the top of the domain to particle mass flux through planes with varying $z$ coordinates. Each line represents the mass flux through an annular region with a lower radius described in the key. The size of particle for this simulation is $300\,\text{nm}$.

between $z$ coordinates (relative to the propeller plane) $0\,\text{m}$ and $0.3\,\text{m}$ with an equal spacing of $0.05\,\text{m}$, in addition to a control plane at the inlet to normalise the results against. The planes were then split into six sampling regions, which were annular in shape with inner radii between $0\,\text{m}$ and $0.25\,\text{m}$ (and a maximum outer radius of $0.3\,\text{m}$). Using the area of these regions and the number of particles passing through them per second, mass flux was derived as a convenient parameter to relate to OPC

measurements. Mass flux was chosen over mass concentration so as to separate the artefacts generated by distortions in the trajectory of the particles, from the artefacts generated by deriving mass concentration from the velocity of the SUA.

The LPT simulations were run for a variety of particle sizes ranging from $300\,\text{nm}$ to $10\,000\,\text{nm}$ (spaced quasi-logarithmically). This size range was chosen because it was predicted that smaller particles would be most affected by erroneous airflow, due to their lower inertia. The results of these simulations are presented in Sect. 4.2.

**4.2    CFD-LPT Results and Discussion**

The simulated mass flux through the annular regions on a series of planes described in Sect. 4.1 is shown in Fig. 6. The plot shows the variation of mass flux with the $z$ co-ordinate of the plane the regions are coincident to (or the elevation above the propellers). The mass flux ratio is the ratio of the mass flux through a region at the top of the domain (therefore unaffected by the propellers) to the mass flux in the same region at a different elevation. The purpose of this simulation was to determine

the best location on the airframe for the OPC instrument. Since the OPC directly measures mass flux, Fig. 6 visualizes the distortion in measurements due to the SUA propellers.

Figure 6 shows a consistently higher mass flux ratio for the annular region in the centre of the domain, meaning less particles are passing through this area. This is because the propellers create a pressure gradient in the continuous phase, and the resulting drag force on the particles pushes them out from the centre. While this drag force is lowest at the centre region (since it is

furthest from the propellers), it still shows consistently lower mass flux than the outer regions because the pressure gradient is



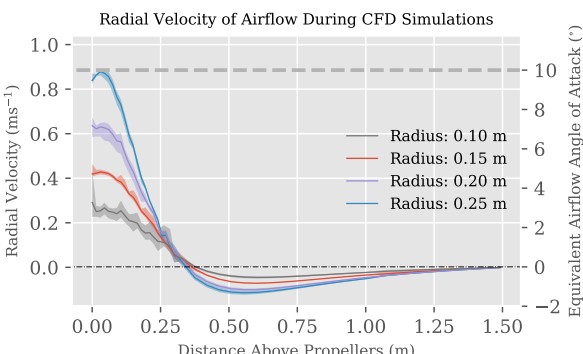

**Figure 7.** CFD results for the (circumferential averaged) radial velocity of the airflow as a function of distance above the propellers. The shaded regions around the lines show the variation of the air velocity around the circumference of the domain, while the bold lines show the mean. The dashed line represents the angle of attack limit specified in Smith et al. (2019).

forcing particles outward towards the propellers and is essentially 'draining' the inner region. This is visualised in Fig. 7, which shows the circumferential average of the radial velocity (implying a radial pressure gradient since the radial velocity should be zero if no other forces are acting on the continuum). Positive radial velocities in this figure mean airflow moving outwards from the centre towards the propellers, and positive radial velocities for all annular regions are shown with $z$ coordinates lower

5  than $0.3\,\mathrm{m}$. Figure 7 also shows the angle of attack of the airflow with respect to the centre ($z$) axis of the domain. The purpose of this is to determine if the direction of the airflow is beyond the angle of attack limit of the UCASS sensor described in Smith et al. (2019) ($10°$). While the airflow angle of attack at all points on this graph was acceptable, a lower 'nominal' angle of attack will make the SUA more robust in crosswinds which will further alter this variable. Additionally, the acceleration of the airflow close to the propellers, and the high levels of turbulence in the continuous phase here mean the droplets are more likely

10  to experience breakup effects. If a larger droplet was broken up into multiple smaller droplets—which were then sampled—the data products would show a shifted size distribution and a larger number concentration. According to Reitz (1987), the breakup conditions are given by

$$We > 6 \tag{1}$$

and

$$We/Re^{0.5} > 0.5 \tag{2}$$



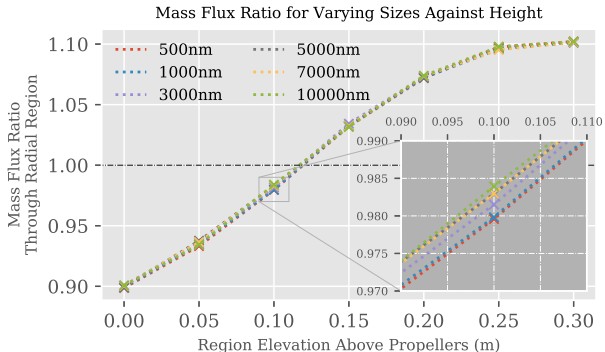

**Figure 8.** CFD-LPT results for the ratio of particle mass flux at the top of the domain to particle mass flux through planes with varying $z$ coordinates. Each line represents the mass flux for a different size particle described in the key. The annular region for this plot has a lower radius of $0.15\,\mathrm{m}$ and an upper radius of $0.2\,\mathrm{m}$.

representing 'bag' and 'stripping' breakup respectively. The Weber number of the droplets and Reynolds number of the flow are given by

$$We = \frac{\rho_1 v_1^2 d_p}{\sigma_2} \tag{3}$$

and

$$Re = \frac{\rho_1 v_1 d_p}{\mu_1} \tag{4}$$

respectively. The subscript $1$ denotes the continuous phase and $2$ the discrete phase. $We$ is Weber number, $Re$ is Reynolds number, $\rho$ is density, $v$ is velocity, $d_p$ is particle diameter, $\mu$ is dynamic viscosity, and $\sigma$ is surface tension.

In both cases, the likelihood of the actual droplet sizes being altered is proportional to the square of the continuum velocity, which is likely to be higher closer to the propellers. Nevertheless, at all points in the analysed regions, neither the limit specified

10 in Eq. 1 nor the limit in Eq. 2 were reached.

Figure 8 is intended to demonstrate the effect the propellers have on the particle size distribution. It was hypothesised that droplets with higher sizes (and hence larger inertia) would be less effected by the propellers. While Fig. 8 shows this is true with the $10\,000\,\mathrm{nm}$ size showing a mass flux ratio consistently closer to unity, the difference between the sizes is extremely small, with a mean standard deviation of 0.00097. This means that the propellers will have a small effect on the shape of

15 the PSD when compared to its peak-height. Not only does this make it easier to apply correction factors to measurements, but atmospheric phenomena, which only require measurements of a normalized PSD (e.g. Zeng, 2018), will be distorted by a negligible amount. Therefore, a correction factor to the particle size distribution is not required.

In contrast to most aerosol sampling methods, the smaller aerosols with less inertia are more problematic in this situation due to the airflow acting to pull particles away from the OPC. However, when a crosswind is introduced, the larger particles





would become more problematic with their greater inertia causing them to impact against the walls of the RCASS. To reduce this artefact, the SUA can be allowed to drift with the wind when taking a vertical sampling profile with a crosswind. Smith et al. (2019) shows that the maximum angle of attack the UCASS can adapt to is $10°$, therefore the SUA will be drifting with the wind whenever a crosswind greater than approximately $1\,\mathrm{m\,s^{-1}}$ is expected, assuming an ascent rate of $5\,\mathrm{m\,s^{-1}}$. This,

however, is not necessary for measurements of radiation fog since this phenomenon is typically associated with wind speeds less than $1\,\mathrm{m\,s^{-1}}$.

Figure 6 shows, for regional elevations greater than $0.15\,\mathrm{m}$ above the propellers, lower regional radii between $0.15\,\mathrm{m}$ and $0.25\,\mathrm{m}$ (corresponding to an annulus with a lower radius of $0.15\,\mathrm{m}$ and an upper radius of $0.3\,\mathrm{m}$) are acceptable. However, Fig. 7 shows larger radial velocities closer to the propellers. While these radial velocities are within the acceptable range for the

UCASS, the addition of a crosswind would add to the maximum radial velocity. Therefore, the RCASS was positioned on the SUA on an annulus with a lower radius of $0.15\,\mathrm{m}$ and an upper radius of $0.20\,\mathrm{m}$, since this region exhibits the lowest average radial velocity.

## 5  Field Validation Experiment

### 5.1  Field Test Method

In order to evaluate the quality of the data collected by the UH-AeroSAM, a practical test was devised involving the comparison of SUA data with a calibrated in-situ cloud probe. While conventionally the validation of simulation data is achieved using a wind tunnel and measurements of a scalar quantity (for example pressure), the size of UH-AeroSAM makes it difficult to find an appropriate tunnel, accounting for a mandated blockage ratio and wall distance. Also, a spatially homogeneous particle stream with a known size distribution in the tunnel is difficult to achieve, and various environmental factors (for example

turbulence) are hard to simulate in a tunnel.

Since RCASS is an OPC, and thus accurate sizing of particles relies on a known refractive index and assumes a spherical shape, the material chosen for the validation was water droplets in low level stratus (and fog). This minimised any uncertainty originating from an unknown particle refractive index; particles with unknown shape or non-homogeneous structures; or unknown artefacts resulting from surface roughness. Since both the RCASS and the reference instrumentation are OPCs, it may

initially appear that these uncertainties would apply equally to both. However, different optical designs will capture different parts of a particle's phase function (most OPC cloud probes, including the CAPS, use forward scattering), which will not vary proportionally to each-other with particle size.

The validation experiments were undertaken during the Pallas Cloud Experiment (PaCE) at the Pallas atmosphere-ecosystem super-site. This is located 170 km north of the Arctic Circle (67.973°N, 24.116°E), partly in the area of Pallas-Yllästunturi

National Park (Lohila et al., 2015). A combination of a temporary restricted airspace (D527 - Pallas), a common occurrence of low level (300 to $500\,\mathrm{m}$ ASL or 0 to $100\,\mathrm{m}$ above the SUA operations site) layered cloud, and the Sammaltunturi station made PaCE an ideal setting for the validation of UH-AeroSAM. For the duration of PaCE, a Cloud and Aerosol Precipitation Spectrometer (CAPS, Baumgardner et al. (2001)) from Droplet Measurement Technologies (DMT) was positioned at the



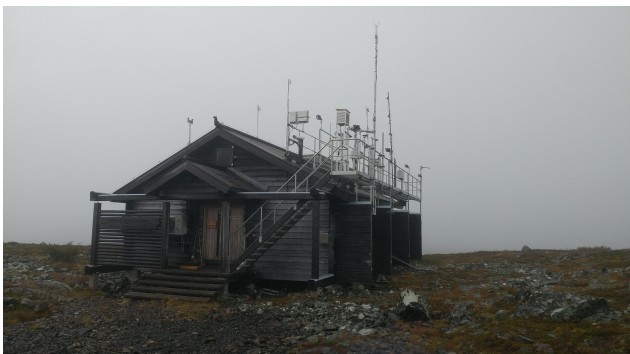

(a) An image of the Sammaltunturi station taken on the 16[th] of September 2019.

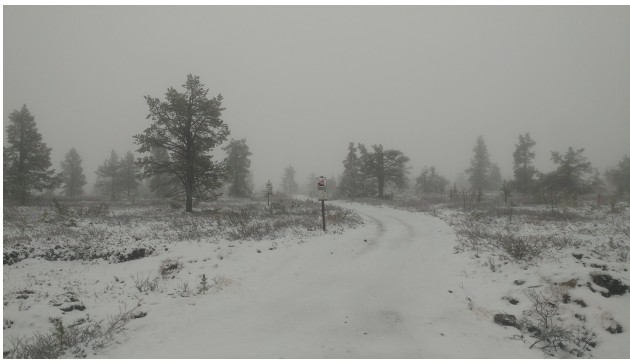

(b) An image of the SUA operations site taken on the 28[th] of September 2019.

**Figure 9.** Images of the Sammaltunturi station and SUA operations site. The Sammaltunturi station is $545\,\mathrm{m}$ ASL, and $100\,\mathrm{m}$ above the SUA operations site.

Sammaltunturi station at the peak of a hill. The CAPS, by default, is a naturally-aspirated instrument which is unsuitable for static measurement, so a pump and inlet system was used to draw a sampling flow and turn it into an artificially-aspirated instrument. Details of this, along with evaluation, can be found in Doulgeris et al. (2020).

Between the 20[th] and 28[th] of September 2019, 24 sampling flights where conducted. However, due to a predominantly

5    northerly wind, the Sammaltunturi station was only in cloud on the 28[th] when 4 flights were conducted, after which all flight batteries were depleted. Table 1 shows a summary of the flights on this day, including time of day (UTC), altitude reached above sea level (ASL), and the RCASS gain mode. The first flight was performed in 'high-gain' mode (measuring smaller sizes). However, following on-site analysis of the data, it was found that a large proportion of the data appeared in the largest size bin, indicating some particles were being undersized due to saturation of the detector. For this reason, the gain was reduced

10    to 'low-gain' mode for the remaining three flights, increasing the lower threshold to $1\,\mu\mathrm{m}$ and the higher threshold to $40\,\mu\mathrm{m}$. The location of the Sammaltunturi station and the SUA operations site is shown in Fig. 9. The wind speed and direction on the



| Flight | Time (UTC) | Altitude (m) | RCASS Gain |
|--------|-----------|-------------|-----------|
| 1 | 09:08:21 | 657 | High |
| 2 | 09:42:55 | 645 | Low |
| 3 | 10:15:56 | 644 | Low |
| 4 | 10:48:57 | 646 | Low |

**Table 1.** A summary of the flights completed by UH-AeroSAM on the 28[th] of September 2019 during the PaCE 2019 campaign. Altitude is the altitude above sea level (ASL) in metres; and UCASS mode is the electrical gain of RCASS. High-gain mode allows measurements of smaller sizes ($0.4\,\mu m$ to $18\,\mu m$); and low-gain allows measurements of larger sizes ($3\,\mu m$ to $40\,\mu m$).

28[th] of September 2019 were $6\,\mathrm{m\,s^{-1}}$ and 240° respectively. The temperature at the station level was $3.6\,°C$, so neither ice nor super-cooled water were expected.

## 5.2 Field Test Results and Discussion

A comparison of normalised number concentration (dN/dlog(Dp)) can be used to assess the performance of a particle instru-
ment, since it provides a size resolved counting efficiency, and some particle loss mechanisms tend to be size dependent. A comparative plot of normalised concentration for the SUA and CAPS is shown in Fig. 10. To reduce the significance of random artefacts, a $20\,s$ arithmetic mean of the CAPS data, and a $40\,m$ arithmetic mean of the RCASS data were taken, centred on coinciding times and altitudes. This averaging period was chosen to cover the same temporal extent. The error margin in concentration for these data is taken to be one standard deviation over the averaged range, this is represented in Fig. 10 by error
bars.

As a general statement, Fig. 10 shows that the RCASS in both gain modes and CAPS agree remarkably well. As outlined in Sect. 1, one aim of this paper is to outline an operational envelope where the SUA-OPC observations can be considered reliable. Therefore the discrepancies that do exist must be, at the very least, explained. The metric used here to compare the two instruments is the re-binned dN/dlog(Dp), using the bin boundaries for the RCASS (since it has the lower resolution).
This metric is indicative of both size and concentration discrepancies. Figure 11 shows the comparative plots for the re-binned data, along with the regression line. Reduced major axis regression (RMAR) as described in Harper (2016) was used since this technique correctly assumes that neither sampling method is perfect. For Fig. 11b, sizes greater than $30\,\mu m$ were neglected because in all cases both the CAPS and RCASS recorded zero concentration, which would give falsely strong correlation.

Figure 10a shows the only data collected in high-gain mode. This mode showed the best agreement with the CAS, especially
for sizes larger than $7\,\mu m$. Figure 11a shows a regression line gradient of 0.908 and a $r^2$ of 0.785 for the re-binned dN/dlog(Dp). However, these data show disagreement in two distinct places. The first is a dip in the CAPS data between $1\,\mu m$ and $2\,\mu m$, which is undetected by RCASS. This is almost certainly due to the lower size-resolution in this region since it is impossible for the 2 bins (centred around $0.6\,\mu m$ and $2\,\mu m$) to capture this feature. The second is a peak in the CAPS data around $5\,\mu m$; similarly the CAPS data coincident to the other three flights on this day also detected this peak. The RCASS has sufficient





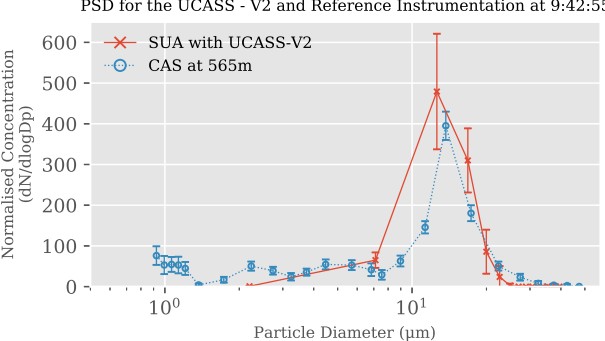

(a) Normalised number concentration against size for the RCASS in high gain mode at 565m above ground level and reference instrumentation at 09:08:21 UTC.

(b) Normalised number concentration against size for the RCASS in low gain mode at 565m above ground level and reference instrumentation at 09:42:55 UTC.

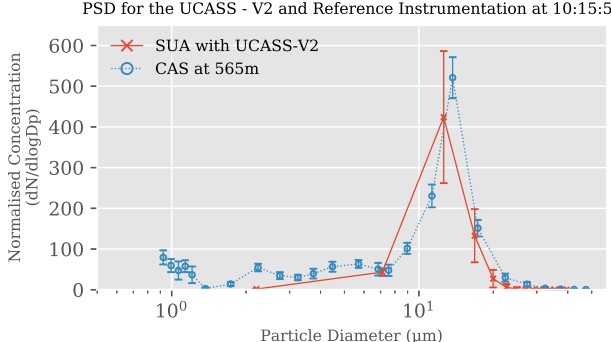

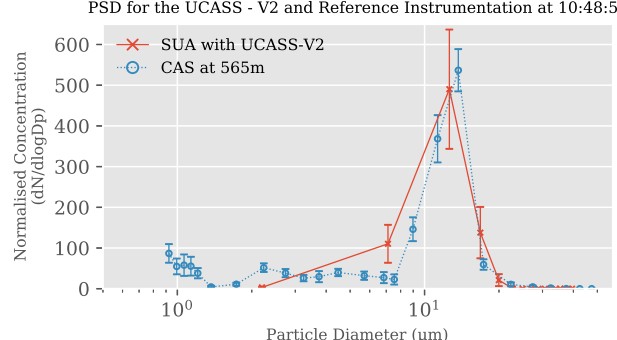

(c) Normalised number concentration against size for the RCASS in low gain mode at 565m above ground level and reference instrumentation at 10:15:56 UTC.

(d) Normalised number concentration against size for the RCASS in low gain mode at 565m above ground level and reference instrumentation at 10:48:57 UTC.

**Figure 10.** Normalised concentration plots for the flights described in Table1. The first flight in Fig. 10a was conducted in high gain mode where it was observed that the largest particle size was beyond the transimpedance amplifier saturation point. The remaining three flights, therefore, were conducted in low gain mode to ensure larger droplets were measured.

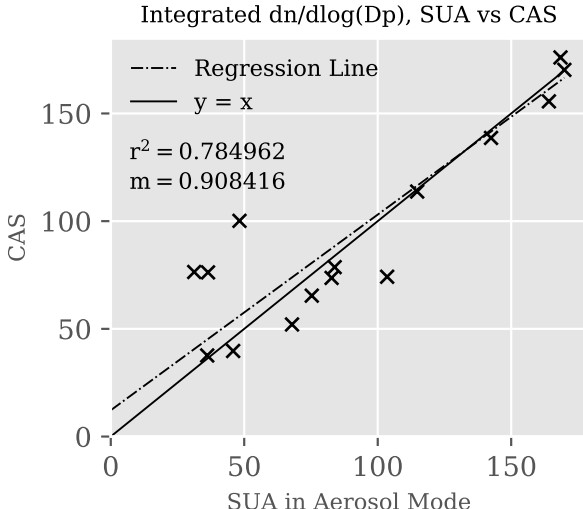

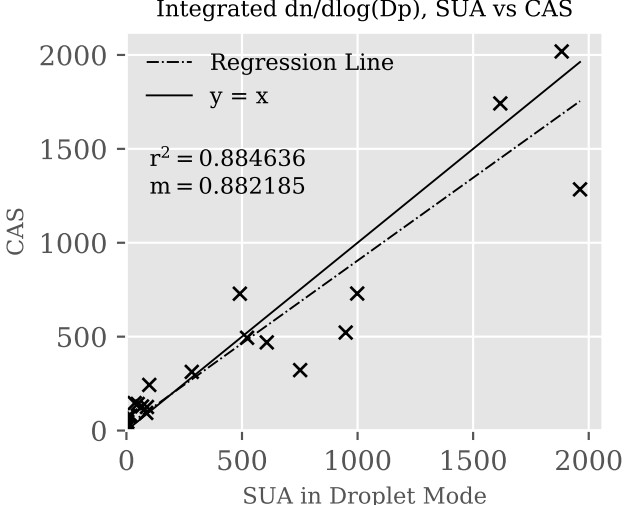

(a) RCASS versus CAPS 1:1 plot of the re-binned data for the flight in high-gain mode.

(b) RCASS versus CAPS 1:1 plot of the re-binned data for the 3 flights in low-gain mode.

**Figure 11.** 1:1 plots for the re-binned dN/dlog(Dp). The solid line represents perfect agreement; the dotted line is the regression line using RMAR. $r^2$ is the coefficient of determination and m is the gradient of the regression line.

resolution to capture this peak only in high-gain mode, but the only flight in high-gain mode did not capture this peak. Within the CAPS user community, the legitimacy of this peak tends to be questioned since it is often not detected by any other instrument. Certainly this could be an artificial double peak due to the highly non-monotonic nature of the Mie curve for a forward scattering instrument; thus RCASS would not detect the peak with the more monotonic 60°-centred (with a half angle

of 44°) scattering angle response curve (Smith et al., 2019). This could also be an artificial peak due to droplet shattering, although the small error margin compared to RCASS would indicate otherwise. Since this is not the topic for this paper, and this peak does not significantly affect the plots in Fig. 10, this part of the graph is not discussed further here.

Figures 10b, 10c, and 10d show all the data collected in low-gain mode. This instrument mode has sufficient range to capture the entire major peak detected by the CAPS; and for the most part the two data sets agree well, although there are a

10 few important points of discussion. The first is a lower resolution for smaller sizes, which causes the instrument to appear as though it disagrees. The worst case of this is presented in Fig. 10b. Since the high-gain mode agrees remarkably well with a higher resolution, and the only change between the two modes is a transimpedance amplifier gain, the issue can be resolved by adjusting the bin boundaries to give more resolution in this area during future measurements.

The second point is a concentration measured close to zero in the first bin. This is likely due to the instruments lower

sensitivity to smaller sizes, characteristic of a typical OPC where the sample volume is defined optically. Since the useful data products that would be calculated from the RCASS low-gain mode, for example liquid water content (LWC), would mostly





rely on larger sizes, this error is insignificant for the most part. Nevertheless, the authors note that one would need to consider this effect when using data from sizes approaching the spectral limits of any OPC. It was considered this effect could be due to smaller particles being pulled away from the propellers, as demonstrated in Sect. 4.2. However, this possibility was dismissed since the RCASS in high-gain mode did not suffer from this artefact.

The third, and most important, artefact is the drop off in the concentration commonly observed by RCASS for sizes greater than $25\,\mu m$, which are particularly prominent in Fig. 10b. This is of upmost importance for parameters which rely on particle volume (for example LWC), since larger particles contribute more significantly to the total particle volume of a distribution than smaller ones. The suspected reason for this was, while the SUA was allowed to drift with the wind, it is expected that a slip velocity exists between the airframe and the airflow. This slip velocity would get smaller in magnitude as the SUA ascends,

assuming no wind shear. However, since the station is only $100\,m$ above ground level, the slip velocity would likely not have been smaller than the $1\,m\,s^{-1}$ limit calculated in Sect. 4.2. In future measurements, to compensate for this effect, the SUA could be pre-programmed to fly to a GPS way-point, calculated so the horizontal speed of the airframe will match the wind speed. Flying upwards at faster speeds would also have the effect of increasing the aspiration efficiency of the instrument, although this might be detrimental to temperature and humidity measurements due to the slow response time of lightweight sensors

introducing a lag in measurement. On the other hand, since this [size drop-off] discrepancy is mostly minute in these data, it could also easily be a co-location error. Larger droplets tend to accumulate near surface level in fog (as would be expected due to gravitational settling), which would explain the larger sizes measured by any instruments at the station. However, a more extensive data set would be needed to confirm this. Despite the minor artefacts, these data were remarkably accurate and precise, given the low instrument and platform cost.

## 20 6 Conclusions

In this paper, a novel design and validation technique for sampling aerosols on a multirotor SUA was presented. This research involved development and production of a bespoke SUA instrumeted platform (UH-AeroSAM). The developmental work using both modelling and lab results found that placement of the particle instrument had a significant influence on the particle flux travelling through its sample volume. Along with the airframe, a bespoke OPC (RCASS) was developed with a naturally aspi-

rated inlet, specifically for use on SUA. This OPC was based on the single-use UCASS (Smith et al., 2019), and incorporated several improvements to the sample airflow, electrical design, and mechanical robustness. The SUA is intended to fly vertically with the OPC inlet pointed upwards, to achieve a vertical profile of atmospheric aerosol or droplets. RCASS can function in two gain modes: High-gain to measure smaller sizes with high resolution; and low-gain to measure larger sizes with lower resolution.

The primary tool used in the design process was computational fluid dynamics with Lagrangian particle tracking (CFD-LPT), this revealed the physical processes affecting the flow of particles through the system. When the airflow is axial with respect to the SUA, the propellers create a lower pressure above the propellers and draw particles towards them. This was shown to effect all sizes measured by the RCASS equally. The CFD-LPT simulations revealed a significantly changing particle flux





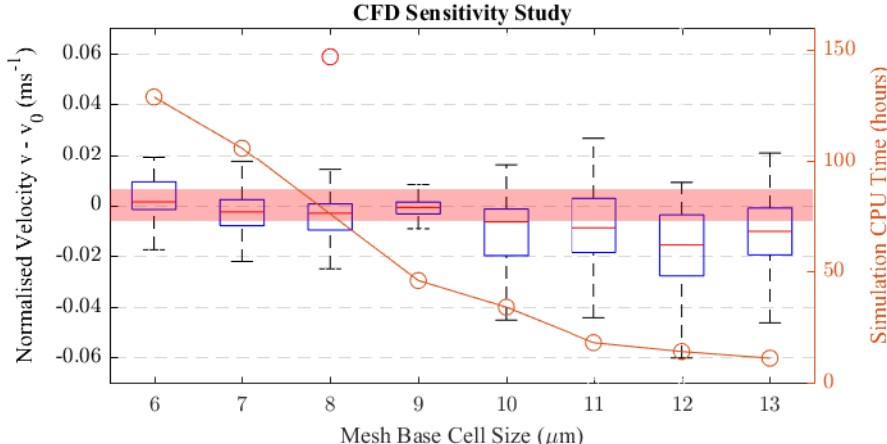

**Figure A1.** A box and whisker plot representation of the CFD sensitivity study results. The whiskers represent the range of the data, the boxes represent the inter-quartile range, and the lines represent the mean. On the left vertical axis is the velocity magnitude normalised to the co-located point in the simulation with the finest mesh size ($5\,\mu m$). The horizontal is the base mesh size. The right vertical axis is the simulation CPU time in hours. The shaded red region represents the valid region for the mean normalised velocity magnitude.

across the airframe due to the propellers, these simulations influenced the positioning of the particle counter on the airframe. An adequately compromised position was found using this method, and the SUA was constructed according to these specifications.

UH-AeroSAM was evaluated in a realistic, field-campaign setting using the cloud and aerosol precipitation spectrometer from Droplet Measurement Technologies as a reference. Across the whole measurement range, the two instruments showed

5   excellent agreement in both RCASS gain modes.

This research delivers hitherto the next step in regular, reliable, and accurate SUA based measurements of particulate and droplet spectra. This is a quantity regularly desired in many fields of atmospheric research, but notoriously difficult to measure on SUA due to the large number of uncharacterised artefacts. However, with the measurement technique presented in this paper, the UH-AeroSAM is capable of accurately measuring the vertically-resolved droplet spectra of fog and low level clouds. This

10  could also be extended to measure particulate matter (PM) concentration, and low level dust loading.

Future research will include adapting the RCASS for fixed wing platforms and characterising the artefacts encountered. While multirotor SUA have better spatio-temporal sampling capabilities, a fixed-wing SUA with a similar maximum take-off mass (MTOM) would have a larger payload mass and a longer endurance. Therefore, it is necessary to explore both applications for different uses.



## Appendix A: CFD Sensitivity Study

To ensure the precision and reliability of simulation results, a mesh sensitivity study was conducted. The CFD domain described in Sect. 4.1 was used without the LPT models, since these are not affected by the mesh. The mesh size was varied between $5\,\mu m$ and $13\,\mu m$, and run for the minimum convergence time, which was found during trials to be 30 time-steps. All simulations were run on the University of Hertfordshire High-Power Cluster (UHHPC) on 64 processors in parallel.

The quantity chosen here to compare the different mesh sizes was the velocity magnitude—spatially averaged across the circumference of circles (concentric with the domain cylinder) with varying radii, and distance above the propellers. The chosen radii ($r$) were 0.1, 0.2, and 0.3 metres, on a plane parallel with the propellers. The distance ($z$) above the propellers was varied up to the top of the domain in intervals of $0.05\,m$. This positioning was chosen because it is the region of interest for the real simulations. This quantity ($|v|_{avg,i}$) was normalized to the co-located $|v|_{avg,0}$ in the simulation with the finest mesh size—$5\,\mu m$—to obtain velocity deviation ($|v|_{avg,i} - |v|_{avg,0}$). The subscript 0 denotes the simulation with the finest mesh size, and $i$ the simulation with mesh size '$i$'. The purpose of the spatial (circumferential) average was to avoid differences in velocity magnitude between time steps due to the position of the propellers, which would exist naturally and give misleading results. The spacing of the circles give 93 values of $|v|_{avg,i} - |v|_{avg,0}$ per simulation. These data, along with CPU time for each simulation, are presented in Fig. A1. The CPU time for the simulation with an $8\,\mu m$ mesh—the red circle in Fig. A1—was dismissed as anomalous due to high UHHPC usage at the time this simulation was conducted

To determine which simulation to use, a limit for the mean velocity deviation had to be established. Here, this limit was defined as the velocity which produces a dynamic pressure capable of moving a particle across $\frac{1}{10}$ of a sampling region $(0.01\,m)$, when applied constantly from the inlet to the propeller plane (for $0.3\,s$ when travelling $1.5\,m$ at $5\,m\,s^{-1}$). This problem can be reduced to one dimension if the pressure is applied constantly as the particle travels in the vertical direction. Therefore the particle must not travel $0.01\,m$ in $0.3\,s$ laterally. The largest mean velocity deviation calculated for a simulation was $\pm 0.0063\,m\,s^{-1}$, using Newtonian equations of motion and assuming Stokes flow. This limit is shown in Fig. A1 as a red shaded valid region.

The coarsest mesh that lies within this limit had a base size of $9\,\mu m$, and was therefore used for the main CFD-LPT simulations.

## Appendix B: Stray Light Test on the RCASS

As an analog to the sun, a 12V halogen bulb with a parabolic directional reflector was used as the stray light source due to its similar spectrum. From previous tests on both the UCASS and RCASS, it was found that stray light had a largest effect with an unobstructed path to the detector. Therefore the RCASS was rigidly mounted with a halogen bulb pointing directly at the detector, corresponding to a solar zenith angle of $25°$ when mounted in the SUA. To measure the equivalent solar power, a laser power meter (calibrated to 500nm, in the middle of the solar spectrum) was used to measure the radiant flux of the halogen light. This was done to make the measurements comparable to true solar irradiance, in order to define a suitable operating envelope for sampling with the SUA. From initial testing, it was found that the RCASS was unaffected by stray light resulting



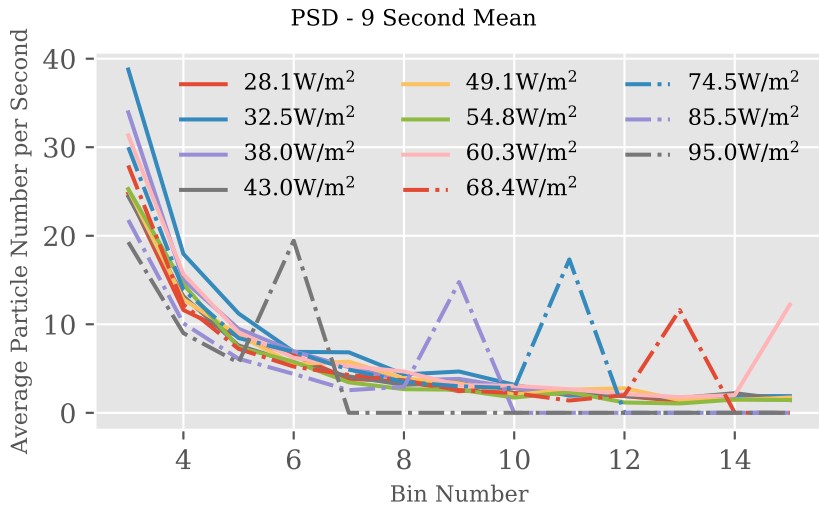

**Figure B1.** The results from the RCASS stray light test. Each line on this graph represents a different irradiance from the halogen bulb used as an analog to the sun. Irradiances larger than this were found to completely saturate the photodiode and cause no counts to be registered.

from a halogen bulb voltage of less then $3.1\,\mathrm{V}$, and completely saturated by bulb voltages more than $4\,\mathrm{V}$. The voltage was therefore varied between $3\,\mathrm{V}$ and $4\,\mathrm{V}$ in steps of $0.1\,\mathrm{V}$, and the irradiance was measured at each data point.

For this experiment, polydisperse water droplets were used as the measurand because of their quasi-spherical shape—so they can be modelled using Mie theory—and ease of generation. A compressed air droplet generator was rigidly mounted

for consistency between experiments. A fan was positioned behind the droplet source, causing a $1.8\,\mathrm{m\,s^{-1}}$ airflow through the RCASS. For each iteration of the experiment, the water droplets were sprayed into the instrument for 9 seconds. The bin boundaries for the test are spaced logarithmically between 0 and 4095 (for the 12-bit ADC).

Figure B1 shows the results from the stray light test. The vertical axis represents the particle counts per second averaged over the 9-second period the droplets were being sprayed, and the horizontal axis is the bin number of the OPC. The different lines

on the plot represent the changing solar-equivalent irradiance. This plot clearly demonstrates the saturation of the photodiode for irradiances larger than $60\,\mathrm{W\,m^{-2}}$. For example, with the halogen bulb at $95\,\mathrm{W\,m^{-2}}$, there is a peak at bin 6, followed by no counts at all for the higher bins. This is because, for the DC signal at $95\,\mathrm{W\,m^{-2}}$, the largest peak possible is 'binned' at a 12-bit ADC value corresponding to bin 6 (578 in this case), and all other particles which would normally—with no stray light—generate larger peaks, are 'binned' here.

Although the effective zenith angular range is small, it was decided post-test that the inlet should be extended so no direct path was available for stray light to reach the detector. This test also revealed that the plastic used for some of the bodywork transmitted infra-red radiation, to which the RCASS detector is highly sensitive, so gold foil was used to surround the inlet exterior and reflect infra-red.





*Author contributions.* The original draft manuscript was prepared by JG and reviewed and edited by all co-authors. Funding was acquired by JG (Aerosol Society travel award). The project was conceptualised by HS, ZU, WS, CS, DC, and JG. Data curation was conducted by JG, DB, and KD. Formal analysis was conducted by JG. Project supervision was the responsibility of HS, WS, ZU, CS, and CC. Methodology was developed by JG, ZU, DB, and RM. Validation was conducted by JG. Software was created by JG and WS.

5   *Competing interests.* The authors declare that they have no conflict of interest.

*Acknowledgements.* We thank the Finnish Meteorological Institute for providing accommodation and logistics throughout the Pallas Cloud Experiment. Travel and shipping was subsidised by a travel award from the Aerosol Society. We also thank the UK Met office Meteorological Research Unit at Cardington for providing support in temperature and humidity instrumentation calibration, and flight testing.



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
