# Peer review of "Design and Field Campaign Validation of a Multirotor UAV and Optical Particle Counter"

_Atmospheric Measurement Techniques, 2020_

## Referee Comment (RC1) · Anonymous Referee #1 · 28 Jul 2020

This is a well written and presented paper and the authors should be commended in tackling the topic of measurement uncertainty arising from particle sensors mounted on UAVs. The UAV platform is an important tool in the atmospheric observer's arsenal but the community has, in general, been lax in applying the same rigour to measurement uncertainty as is applied to sensors on large aircraft platforms.

1. The authors have performed extensive simulations to determine the effects of the propellers and flow through the particle measurement system on the resultant measured size distribution. Particle size is affected by changes in the temperature and humidity of the carrier phase and these parameters can be affected by the flow dynamics. In addition to the ambient field measurements of temperature and humidity, ideally, these would also be measured at the point at which the particle size is measured. As this is difficult to do it is often advisable to model the dynamical effects on these fields in order to get a handle on a possible measurement uncertainty introduced.

2. The authors have been primarily concerned with droplet but there are applications for where the particles under consideration are dry: dust for example. Have the authors modelled the electric field around the particle sensor and considered how this may bias any measurements?

3. The reasoning around using fog and low cloud as the source of "calibrating" particle is sound as these can be assumed to be spherical water droplets (the previous point concerning particle growth due to dynamical effects on temperature and pressure is thus quite important). It is noted that a CAPS probe was used as the reference instrument and of the two optical sensing systems in this probe only the CAS is used (CIP:12.5  $\mu$ m – 1.55 mm (standard), CAS: 0.51  $\mu$ m to 50  $\mu$ m). Given the conditions expected at the deployment site, a DMT FM-120 (ground-based fog monitor: 2  $\mu$ m to 50  $\mu$ m) or CDP-2 (cloud probe that can be used in ground-based applications without the need for artificial aspiration: 2  $\mu$ m to 50  $\mu$ m) would have been a better choice of probe type. Why was the CPAS selected?

4. When using what are essentially aircraft instruments such as CAPS in ground-based deployments it is standard practice to have the instruments automatically aligning so they always point into the wind. Probes such as these have a limited acceptance angle (when deployed on an aircraft they are always pointing directly into the wind by default) and measuremtns can be severely biased when the incident airflow is outside of this. What arrangements were taken to ensure that the ambient wind direction was within the acceptance angle of the CAPS probe?

---

## Referee Comment (RC2) · Anonymous Referee #2 · 13 Aug 2020

General Comments:

This paper is well written and makes a significant contribution to the field. The redesign of the OPC for flight onboard the custom-built SUA is very clever. The greatest contribution of this paper, however, is the rigor with which CFD-LPT simulations are used to evaluating a SUA sampling capabilities and instrument redesign. The authors also do an excellent job of outlining how the custom-built small-unmanned aircraft (SUA) is key to aerosol sampling using the open path optical particle counter (OPC).

As this SUA is a custom built platform, more details, perhaps in a table, on how to set up the CFD-LPT simulations would be helpful, so that future studies may easily emulate

this rigorous theoretical framework for the evaluation of new SUA and instruments. Previous studies that have attempted to use CFD to validate SUA measurements should be cited and discussed briefly, stressing key differences or novel elements of the approach outlined here (e.g. McKinney et al. 2019 AMT).

It would be worth more carefully explaining the scientific value of using the SUA-UCASS-V2 instrument within the bottom 100 m of the boundary layer in particular, compared with other small, light-weight OPC (e.g. how near surface measurements of particles within a wide size range or measurements of liquid cloud water are particularly useful to give the reader an idea of the impact of these measurements and this technology).

The empirical tests, comparisons at high gain of smaller size diameter particles seems somewhat incomplete, particularly when compared with the rest of the paper. Due to the potential artifacts of the CAS at low aerosol sizes, it would be good to see the SUA-UCASS-V2 compared to a different OPC for aerosol sizes between 500 nm and 3.5 $\mu$m, such as POPS, which does not share the same artifacts as CAS either in flight (onboard a tethered balloon to avoid potential influence of aircraft flow), or even, in the lab. Also, could bins not have been reconfigured (either in post processing) or in a second test for a closer comparison in the 1-7 $\mu$m diameter size range, where much of the disagreement between redesigned SUA-UCASS-V2 and CAS instruments exist? There is also considerable disagreement between the OPC high gain (aerosol mode), low gain, and CAS within this size range. The suggestion that small particles could have been influenced by rotor airflow is perhaps too quickly dismissed. As briefly discussed,

Specific Comments:

Throughout change artefact to artifact

P1-L5 Consider rephrasing. It is not clear that the miniaturization of particle instruments hinders accurate and representative measurements, as much as that aerosols

of various sizes may be impacted by the rotor airflow and the SUA's flight patterns during sampling.

P2-L9 Also cite Gao et al. 2016

P2-L22 spatial variation also?

P4 The authors provide two examples of studies that utilized fixed-wing SUA. Because so much of this paper centers on integrating the engineering design with measurement validation of a multi-rotor SUA, it would be better if the discussion could focus primarily on these studies (at least one more example would be good).

P5-L5 What is the endurance (min) of the UH-AeroSAM with a 3.2 kg payload? What does the current payload weigh, if it's not 3.2 kg, and what is the SUA's endurance with this payload?

P5-L13 Under what conditions is the inlet swapped? How do the two inlet configurations differ and why is it helpful or important?

P6-L15 Are raw data saved (and can they be reprocessed)? Are 16 bins the maximum number of bins?

Fig. 9. This figure is not particularly helpful and could be omitted. A diagram showing how the comparison was conducted might be more helpful.

Table 1. This table also seems unnecessary, seeing as all the information within is repeated in Fig. 10. Delete.

Fig 11b. Most of the data is clustered near the origin on this graph – consider a log/log plot here.

[Figure]

---

## Author Comment (AC1) · 2 Sep 2020

Thank you for your comments, here are my responses in order:

1. The induced change in continuous phase temperature changing particle size was considered during the simulation analysis. However, the maximum change in temperature throughout all points considered for sampling—including parts along the trajectory of a droplet from when it enters the domain to when it gets sampled—was 0.012°K. Considering a droplet would be exposed to this temperature field for 0.16s, since the vertical distance this temperature field extends to is 0.8m above the propeller plane, this is certainly not enough of a perturbation to cause any significant change in

droplet size. The manuscript has been amended to reflect this in section 4.2.

2. The electrical field was never modelled, because all the testing was conducted with droplets which carry very little charge and are unlikely to be defected or scavenged by a charged airframe. This deflection artefact, therefore, is not applicable to the paper as it stands. However, your comment does highlight the need to consider such artefacts in the design process when applying these measurements to solid particles, particularly in areas where high charge density is likely to be encountered. In principle CFD-LPT could be combined with electrostatic modelling to characterise particle trajectories in a similar way to how CFD-LPT was used in this paper as it stands. However, the electrostatic portion of the model will be strongly dependant on the charge of an individual particle. This can significantly vary depending on aerosol type—and quantitative analysis of dust turbulent triboelectric charging, for example, is awaiting conclusions of ongoing research (Daskalopoulou et al., 2020). The manuscript has been amended to reflect this in section 4.1.

3. The DMT FM-120 cannot be operated at temperatures below $0°C$, which were expected during the campaign. The CAS overlaps more of the RCASS size range than both the FM-120 and CDP.

4. Thank you for picking this up, the CAPS was oriented into the wind throughout all measurements. I have added a comment in section 5.1 stating this.

References:

Daskalopoulou, V., Mallios, S. A., Ulanowski, Z., Hloupis, G., Gialitaki, A., Tassis, K., and Amiridis, V.: The Electrical Activity of Saharan Dust as perceived from Surface Electric Field Observations in Greece, Atmos. Chem. Phys. Discuss., https://doi.org/10.5194/acp-2020-668, in review, 2020.

---

## Author Comment (AC2) · 2 Sep 2020

Thank you for your comments.

A table has been added in section 4.1 giving better detail behind the CFD-LPT setup. The differences between this study and previous ones are now stressed in the same section. A paragraph in the introduction has been added explaining the significance of measuring cloud droplets in the lower part of the atmosphere.

Since the raw data cannot be saved, see an explanation in an answer to one of your specific points, the bins cannot be reconfigured to have greater resolution in post-

processing. This the why the re-binned data is used for the main comparison in Fig. 11. Focusing on Fig. 11a—since this is the only flight in high-gain mode—the only feature in which there is enough resolution to say the two instruments definitely disagree is the peak around 5-7um, which is often considered an artefact of the CAS. A comparison with POPS is planned, but this will be presented in a future paper.

With regards to your point on dismissing the undercounting of particles due to the airflow effects described in previous sections too quickly, the simulations also show that all the droplet sizes relevant here would be affected by an equal magnitude. Since, as you correctly point out, Fig. 10 shows that the droplet sizes with the highest uncertainty are below 7um, I think it is unlikely that this is the cause. The manuscript has been amended stating this.

In response to your specific comments in order:

1. Artefact is the English (US) spelling and artefact is the English (UK) spelling (https://www.lexico.com/definition/artefact).

2. The abstract has been rephrased to reflect that the miniaturisation of many optical components—for example lasers, optics, and detectors—often leads to lower thermal and electrical stability and (for example photodiodes against photomultiplier tubes).

3. POPS is now cited as an example.

4. The phrase was changed to spatio-temporal as opposed to just temporal.

5. Another example was added to section 2.

6. With a take off mass of 3.2kg the endurance was 13min. The take off mass used was 2kg, the endurance for which is 18min. These figures were obtained from averaging flight logs for flights when the batteries were fully depleted. The manuscript has been changed to reflect this.

7. Thank you for pointing this out. The intention behind this statement was to empha-sise that, since the RCASS and SUA had to be developed in tandem with a tight time constraint, the design contained features which could be changed quickly—such as the inlet—without altering the core design. Since you have pointed this out, however, I realise that there is not much point stating this in a research article, so I have taken it out.

8. The raw data are not saved. This would take too long to transfer to the data-logger and lead to an unacceptable level of sub-sampling at the same temporal resolution, and the high resolution is one of the major benefits of the SUA platform, so it is unwise to compromise this. 16 bins are the maximum and a faster instrument microcontroller would be required to increase this.

9. Fig. 9 has been deleted.

10. Table 1 has been deleted.

11. Fig. 11b now has a log-log scale and the points are easier to distinguish.
* * *

---

## Author Response (AR2)

**Authors Response to Minor Revisions.**

I thank the editor for their further comments and notes. Below are my responses in order:

- I agree that "limited number of" is a better phrase; this has now been changed.
- The sentence now starts a new paragraph. "better" was referring to "better for comparison with LIDAR"; this has been clarified.
- The RCASS acronym is now defined here, and the later definition has been removed.
- Thank you for pointing this out, the statement has been changed to "there exist several benefits" following your comment.
- The position of the table caption in Table 1 has been amended.
- Axes have been added to Fig. 5 with the same labels discussed in the text.
- Figure 6 has been added to clarify the position and dimensions of the LPT sampling regions.
- Figures 9 and 10—now 10 and 11 respectively—have been changed so all the text is in the main figure caption as opposed to the subfigure captions.
- The title of Fig. 10—now Fig. 11—has been changed so the "N" is capitalised in "dN/dlog(Dp)", and "CAS vs SUA". The points are now coloured to represent the droplet size.
- A more detailed description of the operation of the SUA during PaCE was added:

*"In order to ensure quasi-Lagrangian measurements, the UH-AeroSAM was allowed to drift with the wind for all flights. The profiles were, therefore, slanted towards the station due to the 240° wind. The SUA operations site was approximately 450m away 5 from Sammaltunturi station—meaning the SUA was approximately 350m away when its altitude corresponded to that of the station. The RCASS data for comparison were within the altitude range of 545 to 585m ASL, which corresponds to the station altitude ±20m."*

- This statement has been amended to say "fewer" as opposed to "less".
- The comma here is now a semicolon.

[revised manuscript text omitted]